# Molecular diagnosis of *Trichuris trichiura*: Prevalence and associated risk factors in children under five living in a malaria-endemic area in Papua, Indonesia

Dhika Juliana Sukmana[1], Tri Nury Kridaningsih[2], Hartalina Mufidah[3], Enny Kenangalem[4], Faustina Helena Burdam[4], Tri Nugraha Susilawati[5], Jeanne Rini Poespoprodjo[4,6,7*‡], E. Elsa Herdiana Murhandarwati[8‡]

1 Medical Laboratory Technology Program, Poltekkes Kemenkes Yogyakarta, Mantrijeron, Yogyakarta, Indonesia, 2 Balai Laboratorium Kesehatan Masyarakat Papua, Jalan Otonom, Wahno, Kecamatan Abepura, Kota Jayapura, Indonesia, 3 Medical Laboratory Technology Program, Universitas Dr. Soebandi Jember, Jember, Indonesia, 4 Timika Malaria Research Facility, Papuan Health and Community Development Foundation, Timika, Papua, Indonesia, 5 Faculty of Medicine, Universitas Sebelas Maret, Surakarta, Indonesia, 6 Centre for Child Health-PRO, Faculty of Medicine, Public Health and Nursing, Universitas Gadjah Mada, Yogyakarta, Indonesia, 7 Mimika District Hospital and District Health Authority, Timika, Papua, Indonesia, 8 Department of Parasitology, Faculty of Medicine, Public Health and Nursing, Universitas Gadjah Mada, Yogyakarta, Indonesia

* didot2266@yahoo.com
‡ These authors are joint senior authors on this work.

## Abstract

### Background

*Trichuris trichiura*, a soil-transmitted helminth (STH), infection is highly prevalent in children and, if not treated, can cause adverse health consequences. Microscopy using the Kato-Katz method is the mainstay of STH diagnosis in most settings but has low sensitivity. This study aims to quantify prevalence and examine risk factors of trichuriasis among children in a malaria-endemic area.

### Methods

The study recruited 181 children aged <5 years old from a household survey conducted in 16 villages in Timika, Papua, Indonesia, from April to July 2013. Clinical and laboratory data were collected, and stool samples were analysed later using quantitative PCR (qPCR).

### Results

The prevalence of *T. trichiura* infections was 13.8% (25/181; 95%CI, 9.1–19.7) by Kato-Katz microscopy examination and 31.5% (57/181; 95%CI, 24.8–38.8) by qPCR analysis, increasing the prevalence by 17.7% (p = 0.0001). Children aged >3 years old had a 3 times increased risk of having trichuriasis compared to younger

**Data availability statement:** All relevant data are within the paper and its Supporting Information files.

**Funding:** The study was supported by the National Health and Medical Research Council of Australia (Program Grants 1037304) via Professor Ric Price (MSHR, Australia) and Final Project Recognition Grant (grant no.: 3149/UNI/DITLIT/DIT-LIT/LT/2019) from Universitas Gadjah Mada awarded to Professor Elsa Herdiana Murhandarwati.

**Competing interests:** The authors have declared that no competing interests exist.

ones (aOR 3.29 [95%CI, 1.42–7.63], p = 0.006). Children with Hb ≤ 10 g/dL were 2 times more likely to have trichuriasis compared to those with Hb > 10 g/dL, (OR 2.46 [95%CI, 1.14–5.31], p = 0.020). Malaria prevalence was 9.9% (18/181; 95%CI, 6–15.3). Coinfections with malaria increased the risk of anaemia (OR 11.7 [95%CI, 2.0–67.0], p = 0.004. No apparent association was found between trichuriasis and undernutrition (wasting and stunting).

## Conclusions

The prevalence of trichuriasis in under-five children is underestimated and together with malaria, the infections are associated with anaemia. Public health strategy should include STH prevention targeted to young children living in malaria-endemic areas.

---

## Introduction

*Trichuris trichiura* infection, or trichuriasis, is one of the most prevalent soil-transmitted helminth (STH) infections and affected 500 million people globally, with the highest burden being in children [1,2]. Trichuriasis in under-five children is associated with anaemia, malnutrition, stunted growth, and cognitive impairment [1,3,4]. The true burden of trichuriasis in children under five years old is probably underestimated because of the non-specific symptoms of the infection and the use of diagnostic methods that rely on direct microscopy, which has low sensitivity [1,5]. In addition, the chronic nature of the infection as well as the difficulties of treating trichuriasis make it a major public health problem [1]. Although remains inconclusive, another important factor is that STH infections may increase susceptibility to *P. falciparum* infection [6–8]. In a malaria-endemic area in Papua, Indonesia, where STH infections are also prevalent [9], defining the health impact attributable to *T. trichiura* infection would inform strategy to treat and control trichuriasis and other STH infections in this region.

The study aimed to estimate the prevalence of *T. trichiura* infection by using a more sensitive standardized molecular method; i.e., quantitative real-time polymerase chain reaction (qPCR) [10], and to identify risk factors associated with the infection in under-five children living in malaria-endemic areas in Papua, Indonesia.

## Methods

### Study site

The District of Mimika is located in the Papua Province, Indonesia, covering both highlands and lowlands areas. The majority of the population resides in the lowland areas. Malaria is highly endemic in this region, with the annual parasite incidence of 450 per 1000 population at risk in 2013 (at the time of the study), and both *Plasmodium falciparum* and *P. vivax* malaria are similarly prevalent [11]. At the time of the study, precise data on soil-transmitted helminthiases in this region was lacking and the implementation of STH prevention programs was inconsistent, with no iron supplementation programs for children.

## Study population and study design

The study was part of a larger household survey in Timika (District of Mimika) carried out between April and July 2013 in 16 villages in 3 sub-districts as described previously [9,12]. Children aged <5 years old with complete clinical data and stool samples were included in the analysis. Clinical and laboratory data from the 2013 household survey were de-identified and accessed on the 16th January 2019. After collection, stool samples were immediately examined by direct microscopy using the Kato-Katz technique as described previously [9,13]. The remaining stool samples were stored without further processing in a $-80^0$C freezer at the research laboratory in Timika. The stool samples were transferred to Yogyakarta in May 2018 and were transported in a $-20^0$C cool box. Frozen stools were thawed at room temperature and then washed with methods described previously [14]. Directly frozen stool samples, even without preservatives, remain suitable for qPCR detection [15]. Extended freezing preserves DNA integrity, allowing helminth DNA to be effectively recovered during extraction [16,17]. Peripheral parasitaemia was assessed by two trained microscopists and haemoglobin concentration was measured using an electronic Coulter counter (Coulter JTTM, USA). The methods are described in detail elsewhere [9].

## Molecular analysis

Between January and May 2019, stool samples of the children collected during the survey were examined by using singleplex qPCR at the Biochemistry Laboratory (Faculty of Biology, Universitas Gadjah Mada, Yogyakarta). Master mix was prepared at the Parasitology Laboratory, Faculty of Medicine, Public Health and Nursing, Universitas Gadjah Mada, Yogyakarta (see Supplementary File 1 for details). The PCR plates to be run were sealed tightly and placed in an ice box to maintain optimal DNA conditions and further reduce the risk of contamination during the transport from the Parasitology Laboratory to the Biochemistry Laboratory.

## DNA extraction

Stool samples stored at $-20$°C were thawed to room temperature prior to processing. To remove debris and contaminants, samples were washed with 1X phosphate-buffered saline (PBS). Approximately 150 mg of each stool sample was transferred into a 15 mL Falcon tube containing 10 mL of 1X PBS. The mixture was homogenized by vigorous shaking to ensure thorough washing. The tubes were then centrifuged at 2000 g for 3 min, and the supernatant was discarded. This washing step was repeated twice to obtain a clean pellet suitable for DNA extraction [14]. Nucleic acids were extracted using the Quick-DNA™ Fecal/Soil Microbe Miniprep Kit (Zymo Research, Irvine, USA), following the manufacturer's protocol with a minor modification: grinding beads were added, and the samples were processed in a bead beater at maximum speed for 5 minutes. See Supplementary File 2 for the full DNA extraction protocol.

## Singleplex qPCR

Real-time qPCR was performed using a CFX96 Real-Time PCR Cycler (Bio-Rad Laboratories; Hercules, USA). A singleplex real-time qPCR was conducted, starting with the preparation of controls and PCR reagents. A plasmid standard for *Trichuris* served as the positive control. This control was serially diluted from $10^{-2}$ to $10^{-7}$ using nuclease-free water, and amplification was performed to generate a standard curve, which was used to determine the detection limit and to assess positive and negative samples [18]. Nuclease-free water was used as a negative control. The master mix was prepared based on prior optimization and pipetted at 5 µL into 1.5 mL Eppendorf tubes, followed by the addition of 2 µL of homogenized DNA template [19–21]. The thermal cycling conditions were as follows: pre-denaturation at 95°C for 3 minutes, followed by 40 cycles of 95°C for 10 seconds (denaturation) and 61°C for 1 minute (annealing) [14,22].

## Amplification by real-time PCR

DNA amplification by real-time PCR was performed using a pair of forward and reverse primers along with a FAM-labelled probe. The sequences are provided in Table 1 [14,22].

**Table 1. Primer and probe used for *Trichuris trichiura* amplification [14,22].**

| Target | Primer/probe | Nucleotides |
|---|---|---|
| *Trichuris trichiura* | *TrichurisTsmithFor* | 5'GGC GTA GAG GAG CGA TTT 3' |
| | *TrichurisTsmithRev* | 5' TAC TAC CCA TCA CAC ATT AGC C 3' |
| | *TrichurisTwillPr-FAM* | FAM labeled<br>/56-FAM/TT TGC GGG C/ZEN/G AGA<br>ACG GAA ATA TT/3IABkFQ/ |

### Assessment of trichuriasis, nutritional status, and socioeconomic status

*T. trichiura* infection was defined as positive qPCR results using a cycle threshold (Ct) cut-off of 40 based on a previously published study using a similar method [13]. The intensity of infections was obtained from microscopy data (Kato-Katz method) by calculating the number of eggs per gram (EPG) of faeces according to the guidelines as described previously [9,23,24]. Nutritional status was assessed according to the World Health Organization (WHO) 2007 Standard Reference [25]. Wasting and severe wasting were defined as weight-for-height Z-scores below −2 SD and −3 SD, respectively. Similar classification applied for stunting and severe stunting by examining Z-scores of height for age [25]. Households were classified into socioeconomic (SES) groups using Discriminant Analysis of Principal Components based on asset ownership, and then ranked from poorest to richest by both income and expenditure [26]

### Statistical analysis

The questionnaire and laboratory data of the survey were entered into EpiData 3.02 software (EpiData Association, Odense, Denmark). Additional data on stool PCR were added to the database. Data were analysed using SPSS version 25.0 for Windows software (IBM SPSS statistics). Potential risk factors for trichuriasis included for the analysis were age (≤ 3 years old and >3 years old), sex (male and female), ethnic groups (Papuan and non-Papuan), nutritional status by weight for age (normal, wasting, and severe wasting), stunting (yes/no), malaria (yes/no), anaemia (Hb ≤ 10 g/dL and Hb > 10 g/dL) and socioeconomic status (lowest, lower middle, middle, upper middle, and richest). Categorical data were compared by using odds ratios (OR) with 95% confidence intervals (CIs), the chi-squared test with Yates' correction, or the Fisher's exact test. Multiple logistic regression was used to analyse independent risk factors for trichuriasis by entering all potential risk factors.

### Ethical approval

Ethical approval for this study was obtained from the Medical and Health Research Ethics Committee, Faculty of Medicine, Universitas Gadjah Mada, Yogyakarta, Indonesia (KE/FK/1248/EC/2018). Additional informed consent was not required, as prior consent from the 2013 household survey included approval for further analysis of stool specimens. Written consent was obtained at the time, including parental consent for children under 14. The ethics committee approved the study and waived the need for new consent

## Results

### Participants' characteristics

Of 629 children aged <5 years old enrolled in the household survey, 181 had complete clinical data and stool samples for molecular analysis (Fig 1).

The majority of participants were ≤3 years old (142/181, 78.5%) and were of non-Papuan ethnicity (120/181, 66.3%). Nearly one-third of the children had low nutritional status (28.6%, 51/178) and 41% (74/181) were classified as stunted.

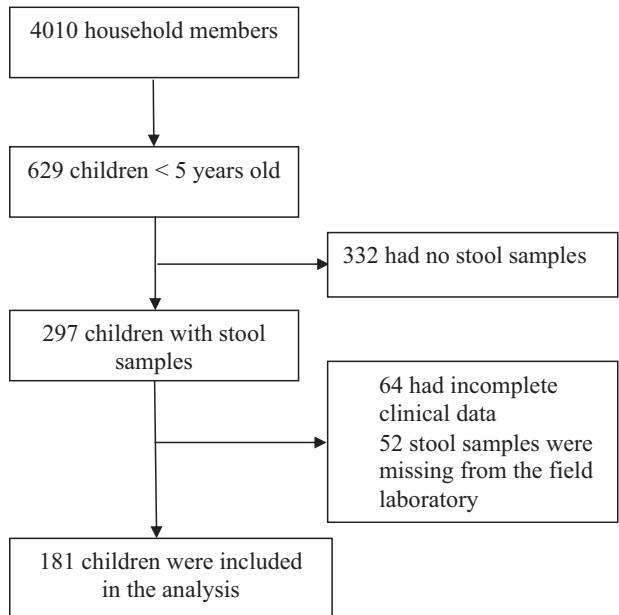

**Fig 1. Study profile.**

Anaemia was found in 18% (33/181; 95%CI, 12.9–24.6) of the children. Malaria prevalence was 9.9% (18/181; 95%CI, 6–15.3) with the majority of infections due to *P. vivax* (Table 2).

## Prevalence of *T. trichiura* infection

The prevalence of *T. trichiura* infection was 13.8% (25/181; 95%CI, 9.1–19.7) by Kato-Katz microscopy examination and 31.5% (57/181; 95%CI, 24.8–38.8) by qPCR, increasing the prevalence by 17.7% (95%CI, 9.1–26.1, p = 0.0001). Of 156 children with negative results by microscopy, 21% [33] had *T. trichiura* detected by qPCR. There was one sample with *T. trichiura* detected by microscopy but not detected by qPCR even after a rerun (Table 3). The range of quantification cycle/cycle threshold (Ct) value observed in this study was from 24.09 to 37.86 and considered as positive for *T. trichiura* infection [13,15].

The intensity of infection in this study was determined based on Kato-Katz microscopic results by calculating the EPG and categorized according to the WHO criteria [27]. Of 25 children with *T. trichiura* infections by microscopy, 84% [21] had mild infections (1–999 EPG) and 4 children had moderate infections (>1000 EPG). Of 25 children with *T. trichiura* infection detected by microscopy, coinfections with *Ascaris lumbricoides* and hookworm were detected in 64% [16] and 32% [8] of the children, respectively (detected by microscopy). Six children had all 3 STH infections.

## Risk factors associated with trichuriasis

Children aged >3 years old had a 3 times increased risk of having trichuriasis compared to younger children (aOR 3.29 [95%CI, 1.42–7.63], p = 0.006). Children with Hb ≤ 10 g/dL were 2 times more likely to have trichuriasis compared to those with Hb > 10 g/dL, (OR 2.46 [95%CI,1.14–5.31], p = 0.020). However, after adjusting for other risk factors, the association was no longer statistically significant (aOR 2.29 [0.92–5.70], p = 0.076). Children of Papuan ethnicity, lower socioeconomic status, and those with severe wasting and stunting were more likely to have trichuriasis, but it was statistically not significant (see Table 4). Malaria was not associated with an increased risk of having trichuriasis.

**Table 2. Characteristics of study participants (N = 181).**

| Variable | Freq (n) | % |
|---|---|---|
| **Sex** | | |
| Male | 96 | 53 |
| Female | 85 | 47 |
| **Age group (years)** | | |
| ≤3 | 142 | 78.5 |
| >3 | 39 | 21.5 |
| **Ethnic groups** | | |
| Papuan | 61 | 33.7 |
| Non-Papuan | 120 | 66.3 |
| **Nutritional status (height for weight)*** | | |
| Normal | 127 | 71.4 |
| Wasting | 28 | 15.7 |
| Severe wasting | 23 | 12.9 |
| **Stunting** | | |
| No | 107 | 59.1 |
| Yes | 74 | 40.9 |
| **Malaria** | | |
| Negative | 163 | 90.1 |
| *P. falciparum* | 3 | 1.6 |
| *P. vivax* | 15 | 8.3 |
| **Anaemia status** | | |
| No (Hb > 10 g/dL) | 148 | 81.8 |
| Yes (Hb ≤ 10 g/dL) | 33 | 18.2 |
| **Socioeconomic status** | | |
| Lowest | 6 | 3.3 |
| Lower-middle | 27 | 14.9 |
| Middle | 15 | 8.3 |
| Upper-middle | 62 | 34.3 |
| Richest | 71 | 39.2 |

*3 missing weight data (excluded from the analysis).

**Table 3. *T. trichiura* detection by method.**

| Kato Katz | qPCR | | Total |
|---|---|---|---|
| | Negative | Positive | |
| Negative | 123 | 33 | 156 |
| Positive | 1 | 24 | 25 |
| **Total** | 124 | 57 | **181** |

## Malaria and trichuriasis coinfection

Of 57 children with trichuriasis by qPCR, the risk of anaemia is significantly increased (75%, 6/8) if co-infected with malaria compared to those with mono *T. trichiura* infection (20.4%, 10/49) with an OR of 11.7 (95%CI, 2.0–67.0), p = 0.004 (see Supplementary File 3). Although the number of coinfection cases is small, the results suggest an adverse haematological consequence of malaria-trichuriasis coinfection.

**Table 4. Risk factors associated with *T. trichiura* infection.**

| Variable | *T. trichiura* infection by PCR, % (n/valid cases) | Univariate Analysis | | Multivariate Analysis | |
|---|---|---|---|---|---|
| | | OR (95% CI) | p | AOR (95% CI) | p |
| **Age group (years)** | | | | | |
| ≤ 3 | 26.8 (38/142) | Reference | | Reference | |
| > 3 | 48.7 (19/39) | 2.60 (1.25-5.39) | 0.010 | 3.29 (1.42-7.63) | 0.006 |
| **Sex** | | | | | |
| Male | 32.3 (31/96) | Reference | | Reference | |
| Female | 30.6 (26/85) | 0.92 (0.49-1.73) | 0.805 | 0.86 (0.42-1.77) | 0.685 |
| **Ethnic groups** | | | | | |
| Non-Papuan | 26.7 (32/120) | Reference | | Reference | |
| Papuan | 41.0 (25/61) | 1.91 (0.99-3.66) | 0.050 | 0.51 (0.24-1.11) | 0.085 |
| **Nutritional status (height for weight)** | | | | | |
| Normal | 31.5 (40/127) | Reference | | Reference | |
| Wasting | 14.3 (4/28) | 0.36 (0.12-1.11) | 0.076 | 0.36 (0.11-1.21) | 0.098 |
| Severe wasting | 47.8 (11/23) | 1.99 (0.81-4.90) | 0.132 | 2.31 (0.79-6.79) | 0.127 |
| **Stunting** | | | | | |
| No | 29.9 (32/107) | Reference | | Reference | |
| Yes | 33.8 (25/74) | 1.19 (0.63-2.26) | 0.581 | 0.98 (0.44-2.18) | 0.960 |
| **Malaria status** | | | | | |
| Negative | 30.1 (49/163) | Reference | | Reference | |
| Positive | 44.4 (8/18) | 1.86 (0.69-5.00) | 0.212 | 2.16 (0.63-7.41) | 0.220 |
| **Anaemia** | | | | | |
| Hb > 10 g/dL | 27.7 (41/148) | Reference | | Reference | |
| Hb ≤ 10 g/dL | 48.5 (16/33) | 2.46 (1.14-5.31) | 0.020 | 2.29 (0.92-5.70) | 0.076 |
| **Socioeconomic status** | | | | | |
| Richest | 29.6 (21/71) | Reference | | Reference | |
| Upper-middle | 29.0 (18/62) | 0.97 (0.46-2.05) | 0.945 | 0.31 (0.03-3.62) | 0.352 |
| Middle | 40.0 (6/15) | 1.58 (0.50-5.02) | 0.432 | 1.03 (0.32-3.29) | 0.961 |
| Lower-middle | 37.0 (10/27) | 1.40 (0.55-3.56) | 0.479 | 2.17 (0.58-8.10) | 0.248 |
| Lowest | 33.3 (2/6) | 1.19 (0.20-7.01) | 0.847 | 1.09 (0.47-2.52) | 0.849 |

## Discussion

This is the first study in this region to demonstrate the prevalence of *T. trichiura* infection detected by qPCR. The prevalence significantly increased by two-fold if detected by qPCR (31.5% vs 13.8% by microscopy), suggesting the undetected burden of trichuriasis in this region. In resource-limited settings, laboratory diagnosis of STHs relies on direct microscopy to detect the presence of eggs in stools by using the Kato-Katz method which has low sensitivity, particularly in low-intensity infections [5,28]. STHs detection by qPCR can increase the sensitivity of diagnosis, resulting in higher prevalence; however, the examination cannot be done as part of routine health care due to high cost (high-tech equipment and PCR consumables) and the need for skilled laboratory personnel [16,28–30].

Limitations of using direct microscopy for helminth egg detection have been well described. The microscopy should be read by trained and skilled personnel which is becoming less common. Another limitation is that the unequal distribution of eggs in a single stool sample (within-sample variations) and daily fluctuations in egg excretion (between-sample variations) may lead to inaccurate results [28,31].

The intensity of the infection can also be examined by using molecular methods either by converting the Cq value or through direct post-qPCR quantification, provided that the absolute concentration of the standard is known [29,32]. However, in this study, the DNA concentration measured by NanoDrop (0.3–5.2 ng/µL) did not show a correlation with the corresponding Ct values. As a result, our findings were interpreted qualitatively. In addition, possible DNA degradation due to long term sample storage may lead to underestimation of the true prevalence and may hinder the assessment of infection intensity [15,16].

*T. trichiura,* like other STHs, is transmitted via the oral-fecal route and is closely associated with poor hygiene and sanitation practices and limited access to clean water [5,33]. The highest burden is in children aged 5–15 years of age, who generally have higher exposure risk to soil-contaminated environments due to their exploratory behaviour [5,34–36]. In this study, older children aged >3 years old are three times more likely to have trichuriasis compared to those aged ≤ 3 years old.

Children ≤ 3 years old are generally under parental/caregivers' supervision and tend to be less mobile compared to older children and thus have a lower risk of infection. However, it has been reported that children under five years old are also at risk of STH infections as early as 1 year old [37]. In this present study, the risk of having trichuriasis starts at an early age, with 27.5% of children ≤ 3 years old being infected and the youngest case is a one-month-old infant. The presence of STHs infection very early in life reflects the poor hygiene and sanitation practices in the household, including among caregivers [5,38,39]. Although poor hygiene and sanitation are associated with low SES, in this study, the risk of infection is similar across all SES. This may suggest limited awareness of STHs transmission and related hygiene and sanitation practices in the study households [27].

This study showed that children co-infected with both malaria and trichuriasis have a higher risk of anaemia compared to those infected with trichuriasis alone. Due to the shared risk factors, malaria and STHs coinfection is common in malaria-endemic areas with an estimated prevalence of 17.7% [40]. The high burden of *T. trichura* infection in early life in malaria-endemic areas is concerning as both infections can cause anaemia, malnutrition, stunted growth, and cognitive impairment; the impact is particularly detrimental in the developing child [1,3,4,9,40,41]. Unlike most STH infections, *T. trichiura* infection tends to be chronic, difficult to diagnose and treat, and more likely to have greater long-term adverse health and growth consequences [1,5].

*T. trichiura* can cause prolonged blood loss due to petechial lesions, mucosal haemorrhage, and active mucosal oozing resulting from worm invasion into the intestinal mucosa and could lead to anaemia [1,5]. Interestingly, although not an independent risk factor, our study showed that children with anaemia (Hb ≤ 10 g/dL) living in a malaria-endemic area were two times more likely to have trichuriasis compared to those with Hb greater than 10 g/dL (OR 2.46; 95%CI, 1.14–5.31; p = 0.020). This finding suggests that anaemia may function as a clinical indicator of underlying trichuriasis in this setting. Iron-deficiency anaemia is known to increase susceptibility to invasive bacterial infections by impairing immune responses [42,43]; however, its association with greater susceptibility to trichuriasis—or other STH infections—remains unclear. In this study, the iron status of the children was not assessed. In a separate analysis, trichuriasis was a significant risk factor for anaemia (OR 2.46 [95% CI: 1.14–5.31]; p = 0.023); however, this association was no longer significant after adjusting for other risk factors in multivariable analysis (see Supplementary File 3).

Coinfection of *T. trichiura* with *A. lumbricoides* and hookworm is also found in young children in this study (all detected by microscopy). Six children had all 3 STHs. The presence of multiple STHs infections in the study population may also contribute to an increased risk of anaemia [9,44]. However, the case number is too small to draw a firm conclusion.

The prevalence of children defined as wasted and stunted in this study is 30% and 41%, respectively, which is relatively high [45,46]. The prevalence of trichuriasis is higher in both children with wasting and stunting, but not statistically significant. The lack of significance may be attributed to the small number of undernourished children in this study. Anaemia, chronic blood loss and impaired intestinal absorption due to inflammation can lead to undernutrition and stunted

growth [1,5,37]. However, assessing causality between STH infection and nutritional status is challenging, particularly in a cross-sectional study [47–49]. In addition, the impact of malaria and STHs on growth is not straightforward and is affected by recurrent or chronic infections along with other factors [46].

This study has several limitations. Firstly, due to funding limitations, we are unable to present STH coinfections based on qPCR. Future studies using multiplex qPCR and more sensitive quantitative techniques could provide important data on the true burden of STH infections in children under five. Secondly, we were unable to fully quantify the validation parameters and therefore used serially diluted plasmid DNA to establish a standard curve. Although this may not fully represent amplification efficiency in study samples, it provided a reliable reference for estimating the detection limit [18]. Lastly, the cross-sectional nature of this study may limit the interpretation of the causal relationship between trichuriasis and anaemia, nutritional status, and malaria. Nevertheless, this study offers preliminary evidence on the underrecognized burden of trichuriasis and its risk factors, highlighting the urgent need to implement routine deworming programs for children under five in this setting.

## Conclusions

The burden of *T. trichiura* infection in children under 5 years of age living in a malaria-endemic area in Papua, Indonesia, is likely underestimated. In this region, trichuriasis in early life is highly prevalent, and together with malaria, the infections are associated with anaemia, a condition predisposing individuals to adverse health, growth, and cognitive consequences. Accurate detection of *T. trichiura* is important; however, qPCR is not readily available in resource-limited settings. In view of this, in addition to malaria control programs, public health strategies should ensure the routine delivery of periodic deworming programs for children under five, along with community education to raise awareness on helminthiasis prevention and promote improved hygiene and sanitation practices in households.

## Supporting information

**S1 File. Trichuris stool PCR Procedures.**
(PDF)

**S2 File. DNA Extraction Procedures.**
(PDF)

**S3 Table. Risk factors for anaemia and coinfection data.**
(PDF)

**S4 File. Discriminant Analysis of Principal Components.**
(PDF)

**S5 File. Timika dataset.**
(XLSX)

## Acknowledgments

We thank the study participants for their significant contribution to the study and the field and laboratory staff of the Papuan Health and Community Development Foundation (PHCDF) in Timika, Papua, Indonesia for their excellent works. We are grateful to Mimika District Health Office, Yayasan Pemberdayaan Masyarakat Amungme Kamoro and Professor Ric Price from Menzies School of Health Research (MSHR – Darwin, Australia) for their continued support. We sincerely thank PHCDF and Professor Ric Price who have kindly shared the stool samples and the study data from the 2013 household survey for the analysis. We are also grateful to Ms. Stacey Llewellyn and Dr. James S McCarthy from the Clinical Tropical Medicine Laboratory, QIMR Berghofer Medical Research Institute, Queensland, Australia for their in kind

contributions (PCR consumables) and for sharing the SOPs, expertise and experiences in doing the molecular analysis. We thank Dr. Jutta Marfurt from MSHR and Dr. Atik Susianto for their kind support in managing the lab works.

## Author contributions

**Conceptualization:** Dhika Juliana Sukmana, Tri Nugraha Susilawati, E Elsa Herdiana Murhandarwati, Jeanne Rini Poespoprodjo.

**Data curation:** Dhika Juliana Sukmana, Tri Nugraha Susilawati, E Elsa Herdiana Murhandarwati, Jeanne Rini Poespoprodjo.

**Formal analysis:** Dhika Juliana Sukmana, Tri Nugraha Susilawati, E Elsa Herdiana Murhandarwati, Jeanne Rini Poespoprodjo.

**Funding acquisition:** Tri Nugraha Susilawati, E Elsa Herdiana Murhandarwati, Jeanne Rini Poespoprodjo.

**Investigation:** Dhika Juliana Sukmana, Tri Nury Kridaningsih, Hartalina Mufidah, Tri Nugraha Susilawati, E Elsa Herdiana Murhandarwati, Jeanne Rini Poespoprodjo.

**Methodology:** Dhika Juliana Sukmana, Tri Nugraha Susilawati, E Elsa Herdiana Murhandarwati.

**Resources:** Tri Nugraha Susilawati, E Elsa Herdiana Murhandarwati.

**Supervision:** Enny Kenangalem, Faustina Helena Burdam, Tri Nugraha Susilawati, E Elsa Herdiana Murhandarwati, Jeanne Rini Poespoprodjo.

**Validation:** E Elsa Herdiana Murhandarwati.

**Writing – original draft:** Dhika Juliana Sukmana, Jeanne Rini Poespoprodjo.

**Writing – review & editing:** Dhika Juliana Sukmana, Tri Nury Kridaningsih, Hartalina Mufidah, Enny Kenangalem, Faustina Helena Burdam, Tri Nugraha Susilawati, E Elsa Herdiana Murhandarwati, Jeanne Rini Poespoprodjo.

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
