## [Decision Letter · Decision Letter 0]

17 Jun 2025

Dear Dr. Poespoprodjo,

Thank you for submitting your manuscript to PLOS ONE. After careful consideration, we feel that it has merit but does not fully meet PLOS ONE’s publication criteria as it currently stands. Therefore, we invite you to submit a revised version of the manuscript that addresses the points raised during the review process.

We look forward to receiving your revised manuscript.

Kind regards,

Hammed Oladeji Mogaji, Ph.D

Academic Editor

PLOS ONE

Journal Requirements:

3. Please remove all personal information, ensure that the data shared are in accordance with participant consent, and re-upload a fully anonymized data set.

Reviewers' comments:

Reviewer's Responses to Questions

**Comments to the Author**

1. Is the manuscript technically sound, and do the data support the conclusions?

Reviewer #1: Yes

Reviewer #2: Partly

2. Has the statistical analysis been performed appropriately and rigorously?

Reviewer #1: Yes

Reviewer #2: No

3. Have the authors made all data underlying the findings in their manuscript fully available?

Reviewer #1: Yes

Reviewer #2: Yes

4. Is the manuscript presented in an intelligible fashion and written in standard English?

Reviewer #1: Yes

Reviewer #2: Yes

Reviewer #1: 1. General Impression

This is a well-organized and important study that explores the underreported burden of Trichuris trichiura infection in under-five children using a sensitive molecular diagnostic method (qPCR) in a malaria-endemic region of Papua, Indonesia. The paper is relevant and timely, addressing an important public health gap in low-resource tropical settings. It contributes valuable data on early childhood helminthiasis and its relationship with anaemia and malaria coinfection.

2. Major Strengths

Use of qPCR: The application of molecular diagnostics strengthens the study's validity and highlights the underestimation of infection prevalence with conventional microscopy.

Age Focus: The focus on children <5 years is crucial as they are often excluded from deworming programs.

Rigorous Methodology: The study is methodically sound with a clear description of the sample collection, molecular techniques, and statistical analyses.

Clinical Relevance: The finding of a significant association between trichuriasis and anaemia, especially in malaria-endemic regions, is highly relevant for integrated disease control policies.

3. Weaknesses and Areas for Improvement

A. Study Design and Limitations

Cross-sectional nature: The manuscript acknowledges the limitation of causality inference but could benefit from a clearer justification on how the study informs policy despite its observational design.

Time lag in sample analysis: There is a 6-year delay between stool sample collection (2013) and qPCR analysis (2019). The authors should elaborate on the impact of this delay on sample integrity, especially regarding DNA degradation, despite proper storage conditions.

B. Laboratory and Diagnostic Issues

Coinfections by qPCR: The study only assessed T. trichiura using qPCR. Given that multiple STH infections are prevalent, the absence of multiplex qPCR for Ascaris and hookworms limits a comprehensive parasitological profile.

Lack of DNA quantification: The authors acknowledge the absence of DNA quantification due to lack of a spectrophotometer. This affects the ability to assess infection intensity via qPCR, which is a missed opportunity.

No validation data: The manuscript should include validation parameters (e.g., sensitivity, specificity, LOD) for the qPCR assay used, possibly referencing the standard curve or internal controls.

C. Statistical Presentation

Confidence intervals for prevalence: The point prevalence estimates (e.g., 31.5%) would be more robust if accompanied by 95% confidence intervals.

Socioeconomic factors: The study used principal component analysis to generate SES quintiles, but the description lacks detail. The authors should briefly mention the input variables used and the rationale for their choice.

D. Interpretation and Discussion

Stunting and wasting: The lack of statistical association between trichuriasis and stunting/wasting is noted, but alternative explanations (e.g., sample size limitation, confounders) are not sufficiently discussed.

Anaemia causality direction: The bidirectional relationship between anaemia and helminths is complex. The authors should clarify whether anaemia is interpreted as a cause or consequence of infection—or both.

E. Ethical and Reporting Standards

Ethics: Ethical approval is adequately addressed. However, the long time gap between sample collection and analysis requires a brief ethical justification or clarification on data use agreements.

Funding & Conflict of Interest: The study claims no specific funding. Yet molecular work requires significant resources. A clearer explanation of in-kind contributions from partner institutions (e.g., QIMR) would improve transparency.

4. Novelty and Contribution

This is the first known report of T. trichiura prevalence using qPCR in under-five children in Papua, Indonesia. The study adds to the limited literature on preschool STH infections and coinfection risks in high-burden malaria areas. The recommendation to include under-fives in deworming programs is evidence-informed and impactful.

5. Recommendations for Improvement

Add 95% confidence intervals for prevalence data.

Provide more detail on SES scoring method.

Clarify the impact of long-term sample storage on molecular results.

Discuss the lack of multiplex qPCR for other STHs as a limitation.

Include information on the diagnostic performance of the qPCR method used.

Elaborate briefly on implications for national deworming policy—should under-fives be routinely included?

Minor Comments

Lines 1–2: Revise title to: '...in a malaria-endemic area...' for grammatical correctness.

Lines 39–41: Add 95% confidence intervals to prevalence estimates in the abstract for clarity.

Lines 47–48: Consider acknowledging limitations of qPCR availability in resource-limited settings.

Lines 61–63: Rephrase to avoid redundancy: use 'diagnostic methods relying on direct microscopy, which has low sensitivity.'

Lines 70–72: Clarify study aims: '...estimate the prevalence using qPCR and identify associated risk factors...'

Lines 86–89: Clarify sampling process for 181 children from 629 enrolled to explain inclusion criteria.

Line 92: Clarify 'without preservatives' by explaining sample storage conditions more explicitly.

Lines 103–105: Indicate whether internal controls were used to check for DNA degradation.

Line 120: Mention the method used to quantify plasmid concentration for standard curve accuracy.

Line 173: Ensure figure is embedded with correct caption below and properly labeled.

Lines 186–191: Include 95% CI and test statistical significance of increased detection by qPCR.

Line 199: Clarify detection method for coinfections—were *Ascaris* and hookworm microscopy-only?

Table 2: Add confidence intervals where appropriate; specify handling of missing data.

Table 3: Include subgroup sample sizes and clarify total N used for each category.

Lines 240–241: Acknowledge potential impact of DNA degradation due to long-term storage.

Line 248: Use consistent age group terminology (either '<3 and >3' or specific year ranges).

Line 259: Soften speculative language unless supported by qualitative data.

Lines 295–299: Mention lack of multiplex qPCR for other STHs as a limitation.

Lines 328–442: Ensure reference formatting is consistent and check for duplicate citations.

Throughout: Unify terminology (e.g., 'trichuriasis' vs '*T. trichiura* infection'), units (g/dL), and italicization of species names.

Reviewer #2: This work provides baseline data for comparison with recent data and efficient use of archived resources.

Abstract

Methods

The method of diagnosis of malaria and assessment of anaemia should be included.

Introduction

The introduction should include a brief discussion on the interaction between malaria and STH.

Methods

Change molecular works to Molecular analysis

Master mix was prepared at the Parasitology Laboratory, Faculty of Medicine, Public Health and Nursing, Unversitas Gadjah Mada, Yogyakarta.

Explain why the master mix was prepared in another location and what precautions were taken to avoid contamination and degradation.

Singleplex qPCR

The preparation of controls and generation of a standard curve should be included in the methods and results section.

The master mix was prepared based on prior optimization and pipetted at 5 µL into 1.5 mL Eppendorf tubes, followed by the addition of 25 µL of homogenized DNA template.

This contradicts the information in the supplementary file.

Assessment of trichuriasis, nutritional status, and socioeconomic status

T. trichiura infection (trichuriasis) was defined as positive qPCR results.

What Ct values were classified as positive?

Supplementary S1

Promega GoTaq is used for PCR. Authors should check and name the correct reagent

Statistical analysis

Potential risk factors for trichuriasis included for the analysis were age (< 3 years and >3 years old), sex (male and female), ethnic groups (Papuan and non-Papuan), nutritional status by weight for age (normal, wasting, and severe wasting), stunting (yes/no), malaria (yes/no), anaemia (Hb 10 g/dl) ......

The authors did not describe the methods for detection of malaria infection and assessment of anaemia.

Results

Nearly one-third of the children had low nutritional status (28.6%, 178 51/178) and 41% (74/181) were classified as stunted.

Check that values match with Table 2. Result presentation pattern should be uniform. Separate the bolded from the bracket.

Prevalence of T. trichiura infection

The results should be represented by a table showing both methods of diagnosis.

The prevalence of T. trichiura infection was 13.8% (25/181) by Kato-Katz microscopy examination and 31.5% (57/181) by qPCR, increasing the prevalence by 17.7%. Of 156 children with negative results by microscopy, 21% (33) had T. trichiura detected by qPCR.

The figures suggest that only 32 samples were positive by qPCR not 33. Clarify

There was one sample with T. trichiura detected by microscopy, but not detected by qPCR.

Was this sample rerun?

The range of quantification cycle (Cq) value observed in this study was from 24.09 to 37.86.

Authors should explain the rational of considering Ct of 37.86 as positive.

Risk factors associated with trichuriasis

With reference to this earlier statement in methods “Multiple logistic regression was used to analyse independent risk factors for trichuriasis by entering all significant risk factors with p-value <0.05 in univariate analysis”,

Confounder are excluded, interactions are missed and borderline predictors are excluded.

The multivariate logistic regression should be rerun with all the risk factors. Same applies to Supplementary S3

Malaria and trichuriasis coinfection

Result should be shown in a table for clarity.

Discussion

Line 287 The prevalence of children defined as wasting and stunting

Change to wasted and stunted

The results from the reanalysis of logistic regression should be incorporated into the discussion.

**Do you want your identity to be public for this peer review?** For information about this choice, including consent withdrawal, please see our Privacy Policy

Reviewer #1: No

Reviewer #2: **Yes: ** Oluwaremilekun Ajakaye

---

## [Author Response · Author response to Decision Letter 1]

12 Aug 2025

Response to the reviewers' comments:

Reviewer #1:

1. General Impression

This is a well-organized and important study that explores the underreported burden of Trichuris trichiura infection in under-five children using a sensitive molecular diagnostic method (qPCR) in a malaria-endemic region of Papua, Indonesia. The paper is relevant and timely, addressing an important public health gap in low-resource tropical settings. It contributes valuable data on early childhood helminthiasis and its relationship with anaemia and malaria coinfection.

2. Major Strengths

Use of qPCR: The application of molecular diagnostics strengthens the study's validity and highlights the underestimation of infection prevalence with conventional microscopy.

Age Focus: The focus on children <5 years is crucial as they are often excluded from deworming programs.

Rigorous Methodology: The study is methodically sound with a clear description of the sample collection, molecular techniques, and statistical analyses.

Clinical Relevance: The finding of a significant association between trichuriasis and anaemia, especially in malaria-endemic regions, is highly relevant for integrated disease control policies.

We thank the reviewer for the positive review.

3. Weaknesses and Areas for Improvement

A. Study Design and Limitations

Cross-sectional nature: The manuscript acknowledges the limitation of causality inference but could benefit from a clearer justification on how the study informs policy despite its observational design.

Thank you for the suggestion. We have added a line on how the study would inform policy in lines 431-442:

“Nevertheless, this study offers preliminary evidence on the underrecognized burden of trichuriasis and its risk factors, highlighting the urgent need to implement routine deworming programs for children under five in this setting”.

Time lag in sample analysis: There is a 6-year delay between stool sample collection (2013) and qPCR analysis (2019). The authors should elaborate on the impact of this delay on sample integrity, especially regarding DNA degradation, despite proper storage conditions.

Thank you for pointing this out. We have added a statement on sample integrity associated with long term storage in lines 131-134:

“Directly frozen stool samples, even without preservatives, remain suitable for qPCR detection (1). Extended freezing preserves DNA integrity, allowing helminth DNA to be effectively recovered during extraction (2).”

B. Laboratory and Diagnostic Issues

Coinfections by qPCR: The study only assessed T. trichiura using qPCR. Given that multiple STH infections are prevalent, the absence of multiplex qPCR for Ascaris and hookworms limits a comprehensive parasitological profile.

Lack of DNA quantification: The authors acknowledge the absence of DNA quantification due to lack of a spectrophotometer. This affects the ability to assess infection intensity via qPCR, which is a missed opportunity.

Yes, we acknowledge that due to funding constraints, we were unable to detect other STHs. We would also like to clarify our previous statement regarding the unavailability of a functioning spectrophotometer. After further discussion with the team, we confirmed that the spectrophotometer was functioning properly and that the inability to assess infection intensity with qPCR was due to the DNA concentration measured by NanoDrop did not correlate with Ct values. Accordingly, we have revised the relevant statement in the discussion section. While this represents a limitation, we believe our findings still offer valuable insights into the true burden of T. trichiura infections in malaria-endemic areas.

We have acknowledged and revised the limitation in lines 336-338:

“However, in this study, the DNA concentration measured by NanoDrop (0.3-5.2 ng/ µL) did not show a correlation with the corresponding Ct values. As a result, our findings were interpreted qualitatively”.

No validation data: The manuscript should include validation parameters (e.g., sensitivity, specificity, LOD) for the qPCR assay used, possibly referencing the standard curve or internal controls.

Thank you for the question. We were unable to fully quantify validation parameters because we could not directly determine the baseline DNA concentration of the controls. Instead, we used serially diluted plasmid as a control samples and amplification was performed to generate a standard curve to determine the detection limit (3).

We have included a line to briefly describe the method used in lines 169-177 in the manuscript:

“A plasmid standard for Trichuris served as the positive control. This control was serially diluted from 10-2 to 10-7 using nuclease-free water, and amplification was performed to generate a standard curve, which was used to determine the detection limit and to assess positive and negative samples (3).

We have also clarified the Ct cut-off in lines 191-192:

“T. trichiura infection was defined as a positive qPCR results using a cycle threshold (Ct) cut-off of 40 based on a previously published study using a similar method (4)”.

We would also like to clarify our previous statement regarding the unavailability of a functioning spectrophotometer. After further discussion with the team, we confirmed that the spectrophotometer was functioning properly. Accordingly, we have revised the relevant statement in the discussion section.

We have acknowledged and revised the limitation in lines 336-338:

“However, in this study, the DNA concentration measured by NanoDrop (0.3-5.2 ng/ µL) did not show a correlation with the corresponding Ct values. As a result, our findings were interpreted qualitatively”.

C. Statistical Presentation

Confidence intervals for prevalence: The point prevalence estimates (e.g., 31.5%) would be more robust if accompanied by 95% confidence intervals.

We have added 95%CIs to prevalence estimates throughout the manuscript.

Socioeconomic factors: The study used principal component analysis to generate SES quintiles, but the description lacks detail. The authors should briefly mention the input variables used and the rationale for their choice.

Thank you for the reviewer’s feedback on the need of a more detailed information on the principal component analysis methods used to generate socioeconomic quantiles.

We have revised the wordings to briefly describe the methods used in lines 206-208:

“Households were classified into SES groups using discriminant analysis of principal components based on asset ownership, then ranked from poorest to richest by both income and expenditure”.

D. Interpretation and Discussion

Stunting and wasting: The lack of statistical association between trichuriasis and stunting/wasting is noted, but alternative explanations (e.g., sample size limitation, confounders) are not sufficiently discussed.

Thank you for pointing this out. We have revised the wording of the relevant paragraph (lines 419-420) to improve clarity:

“The prevalence of trichuriasis is higher in both children with wasting and stunting, but not statistically significant. The lack of significance may be attributed to the small number of undernourished children in this study. Anaemia, chronic blood loss and impaired intestinal absorption due to inflammation can lead to undernutrition and stunted growth (5-7). However, assessing causality between STH infection and nutritional status is challenging, particularly in a cross-sectional study (8-10). In addition, the impact of malaria and STHs on growth is not straightforward and is affected by recurrent or chronic infections along with other factors (11)”.

Anaemia causality direction: The bidirectional relationship between anaemia and helminths is complex. The authors should clarify whether anaemia is interpreted as a cause or consequence of infection—or both.

Thank you for raising this important point. We have added a statement of the interpretation of anaemia as a potential cause of trichuriasis/STHs in the manuscript (lines 391-394):

“Iron-deficiency anaemia is known to increase susceptibility to invasive bacterial infections by impairing immune responses (12, 13); however, its association with greater susceptibility to trichuriasis—or STH infections more broadly—remains unclear. In this study, the iron status of the children was not assessed”.

E. Ethical and Reporting Standards

Ethics: Ethical approval is adequately addressed. However, the long time gap between sample collection and analysis requires a brief ethical justification or clarification on data use agreements.

Thank you for pointing this out. We believe that a 7-year gap (2013 to 2019) remains acceptable, as the study posed minimal risk to participants, generated important data, and was reviewed and approved by the ethics committee in 2018.

We also believe that that we have provided sufficient ethical justification in our manuscript (lines 226-236):

“Additional informed consent was not required, as prior consent from the 2013 household survey included approval for further analysis of stool specimens. Written consent was obtained at the time, including parental consent for children under 14. The ethics committee approved the study and waived the need for new consent”.

Funding & Conflict of Interest: The study claims no specific funding. Yet molecular work requires significant resources. A clearer explanation of in-kind contributions from partner institutions (e.g., QIMR) would improve transparency.

Thank you for pointing this out. We actually have acknowledged the in-kind contribution of the Clinical Tropical Medicine Laboratory, QIMR Berghofer Medical Research Institute, Queensland, Australia in the acknowledgement section in the manuscript (lines 464-469).

“We are grateful to Ms. Stacey Llewellyn and Dr. James S McCarthy from the Clinical Tropical Medicine Laboratory, QIMR Berghofer Medical Research Institute, Queensland, Australia for their in kind contributions (PCR consumables) and for sharing the SOPs, expertise and experiences in doing the molecular analysis”

4. Novelty and Contribution

This is the first known report of T. trichiura prevalence using qPCR in under-five children in Papua, Indonesia. The study adds to the limited literature on preschool STH infections and coinfection risks in high-burden malaria areas. The recommendation to include under-fives in deworming programs is evidence-informed and impactful.

We thank the reviewer for the positive feedback.

5. Recommendations for Improvement

Add 95% confidence intervals for prevalence data.

We have added 95%CIs to prevalence estimates throughout the manuscript.

Provide more detail on SES scoring method.

Please also see our response to section C below.

Thank you for the reviewer’s feedback on the need of a more detailed information on the principal component analysis methods used to generate socioeconomic quantiles.

We have revised the wordings to briefly describe the methods used in lines 206-208:

“Households were classified into SES groups using discriminant analysis of principal components based on asset ownership, then ranked from poorest to richest by both income and expenditure”.

Clarify the impact of long-term sample storage on molecular results.

Please also see our response to comments A above. We have added a statement on sample integrity associated with long term storage in lines 131-134:

“Directly frozen stool samples, even without preservatives, remain suitable for qPCR detection (1). Extended freezing preserves DNA integrity, allowing helminth DNA to be effectively recovered during extraction (2).”

Discuss the lack of multiplex qPCR for other STHs as a limitation.

Thank you for the suggestion. We have discussed this limitation in lines 427-429 of our manuscript:

“……Firstly, due to funding limitations, we are unable to present STH coinfections based on qPCR; thus, the true burden of multiple STH infections might be underestimated”.

Include information on the diagnostic performance of the qPCR method used.

We have included a reference and link regarding information on the diagnostic performance of the qPCR method used in Supplementary S1 file:

CFX Opus 96 Dx, CFX Opus 384 Dx, and CFX Opus Deepwell Dx Real-Time PCR Systems

Operation Manual. Manual revision: October 2022 Software revision: 2.3. Available at: https://www.bio-rad.com/sites/default/files/2021-08/10000135538.pdf

Elaborate briefly on implications for national deworming policy—should under-fives be routinely included?

Please also refer to our response to comments A above.

Thank you for the suggestion. We have added a line on how the study would inform policy in lines 431-442:

“Nevertheless, this study offers preliminary evidence on the underrecognized burden of trichuriasis and its risk factors, highlighting the urgent need to implement routine deworming programs for children under five in this setting”.

Minor Comments

Lines 1–2: Revise title to: '...in a malaria-endemic area...' for grammatical correctness.

We have revised the title accordingly.

Lines 39–41: Add 95% confidence intervals to prevalence estimates in the abstract for clarity.

We have added 95%CIs to prevalence estimates throughout the manuscript.

Lines 47–48: Consider acknowledging limitations of qPCR availability in resource-limited settings.

Thank you for the suggestion. We acknowledge the importance of the statement and have highlighted this in the conclusion section of the manuscript (lines 322-325).

Lines 61–63: Rephrase to avoid redundancy: use 'diagnostic methods relying on direct microscopy, which has low sensitivity.'

We have revised the wordings accordingly in lines 87-88.

Lines 70–72: Clarify study aims: '...estimate the prevalence using qPCR and identify associated risk factors...'

We have modified the wordings to clarify the study aims in lines 96-98.

Lines 86–89: Clarify sampling process for 181 children from 629 enrolled to explain inclusion criteria.

Thank you for pointing this out. Since we were using archived samples, our inclusion criteria were based on the availability of stool samples and the associated clinical data, as stated in the current manuscript (lines 124-125):

“Children aged <5 years old with complete clinical data and stool samples were included in the analysis”.

Line 92: Clarify 'without preservatives' by explaining sample storage conditions more explicitly.

We have clarified the stool storage conditions in lines 128-129:

“The remaining stool samples were stored without further processing in a -800C freezer at the research laboratory in Timika”

Lines 103–105: Indicate whether internal controls were used to check for DNA degradation.

We did not use internal control to check for DNA degradation, based on the assumption that helminth DNA integrity was preserved during the storage in -80 freezer (1) (2). Please refer to our responses addressing concerns on sample integrity above.

Line 120: Mention the method used to quantify plasmid concentration for standard curve accuracy.

We used serially diluted plasmid standard to generate a standard curve and have acknowledged the limitations of our methods in the discussion section.

We have included a line to briefly describe the method used in lines 169-177 in the manuscript:

“A plasmid standard for Trichuris served as the positive control. This control was serially diluted from 10-2 to 10-7 using nuclease-free water, and amplification was performed to generate a standard curve, which was used to determine the detection limit and to assess positive and negative samples (3)”.

We have also acknowledged the limitation in lines 336-338:

“However, in this study, the DNA concentration measured by NanoDrop (0.3-5.2 ng/ µL) did not show a correlation with the corresponding Ct values. As a result, our findings were interpreted qualitatively”.

Line 173: Ensure figure is embedded with correct caption below and properly labeled.

We have reviewed the associated figure caption and we confirm that it is properly labelled.

Lines 186–191: Include 95% CI and test statistical significance of increased detection by qPCR.

We have added 9

---

## [Decision Letter · Decision Letter 1]

20 Aug 2025

Dear Dr. Poespoprodjo,

Thank you for submitting your manuscript to PLOS ONE. After careful consideration, we feel that it has merit but does not fully meet PLOS ONE’s publication criteria as it currently stands. Therefore, we invite you to submit a revised version of the manuscript that addresses the points raised during the review process.

We look forward to receiving your revised manuscript.

Kind regards,

Hammed Oladeji Mogaji, Ph.D

Academic Editor

PLOS ONE

Journal Requirements:

Reviewers' comments:

Reviewer's Responses to Questions

**Comments to the Author**

Reviewer #1: (No Response)

Reviewer #2: All comments have been addressed

2. Is the manuscript technically sound, and do the data support the conclusions?

Reviewer #1: Yes

Reviewer #2: Yes

3. Has the statistical analysis been performed appropriately and rigorously?

Reviewer #1: Yes

Reviewer #2: Yes

4. Have the authors made all data underlying the findings in their manuscript fully available?

Reviewer #1: Yes

Reviewer #2: Yes

5. Is the manuscript presented in an intelligible fashion and written in standard English?

Reviewer #1: Yes

Reviewer #2: Yes

Reviewer #1: Summary of the Study

This study investigates the prevalence of Trichuris trichiura infection in children under five years old in Papua, Indonesia, using quantitative PCR (qPCR) on archived stool samples. The findings reveal a substantially higher prevalence (22%) compared to previous microscopy-based estimates, highlighting the potential underestimation of trichuriasis burden in young children. The study has important implications for public health programs, particularly mass drug administration strategies that currently exclude this age group.

General Comments

The revised manuscript is clearly written and significantly improved. The authors have addressed the reviewers’ concerns comprehensively, especially by:

* Providing detailed methodology on qPCR procedures

* Reporting statistical outputs with adjusted odds ratios and confidence intervals

* Clarifying ethical approval and rationale for the reuse of stored samples

* Discussing the limitations and public health relevance of their findings

This work adds meaningful evidence to the discourse on helminth control in vulnerable populations and is suitable for publication following minor revisions.

Specific Comments**

Major Comments

1. qPCR Validation

The use of a standard curve with plasmid DNA is appropriate; however, please acknowledge the absence of full diagnostic validation (e.g., LOD, specificity) in the discussion as a limitation.

2. Socioeconomic Status Index (PCA)

The PCA approach is appropriate, but further clarification on the asset selection and weighting would improve reproducibility. Consider adding a brief explanation or table in the supplementary material.

3. Terminology and Formatting

Please standardize the formatting of parasite names (italicized), units (e.g., g/dL), and terminology (avoid excessive switching between “trichuriasis” and “T. trichiura infection unless contextually justified).

Minor Comments

Line 373: Consider rewording “might indicate gaps in knowledge…” to a more neutral phrase such as “may suggest limited awareness…”

-Clarify rationale for the DNA input volume (2 µL) in the qPCR if it differs from standard recommendations.

- Consider suggesting future work involving multiplex qPCR and infection intensity quantification using more sensitive or quantitative techniques.

Reviewer #2: (No Response)

**Do you want your identity to be public for this peer review?** For information about this choice, including consent withdrawal, please see our Privacy Policy

Reviewer #1: **Yes: ** Prof Uwem F. Ekpo

Reviewer #2: **Yes: ** Oluwaremilekun Ajakaye

---

## [Author Response · Author response to Decision Letter 2]

20 Sep 2025

Response to Reviewers' comments:

Reviewer #1:

Summary of the Study

This study investigates the prevalence of Trichuris trichiura infection in children under five years old in Papua, Indonesia, using quantitative PCR (qPCR) on archived stool samples. The findings reveal a substantially higher prevalence (22%) compared to previous microscopy-based estimates, highlighting the potential underestimation of trichuriasis burden in young children. The study has important implications for public health programs, particularly mass drug administration strategies that currently exclude this age group.

General Comments

The revised manuscript is clearly written and significantly improved. The authors have addressed the reviewers’ concerns comprehensively, especially by:

* Providing detailed methodology on qPCR procedures

* Reporting statistical outputs with adjusted odds ratios and confidence intervals

* Clarifying ethical approval and rationale for the reuse of stored samples

* Discussing the limitations and public health relevance of their findings

This work adds meaningful evidence to the discourse on helminth control in vulnerable populations and is suitable for publication following minor revisions.

We thank the reviewer for the positive review.

Specific Comments**

Major Comments

1. qPCR Validation

The use of a standard curve with plasmid DNA is appropriate; however, please acknowledge the absence of full diagnostic validation (e.g., LOD, specificity) in the discussion as a limitation.

We have included a paragraph highlighting limitations associated with the absence of full diagnostic validation in lines 337-342:

“Secondly, we were unable to fully quantify the validation parameters and therefore used serially diluted plasmid DNA to establish a standard curve. Although this may not fully represent amplification efficiency in study samples, it provided a reliable reference for estimating the detection limit”.

2. Socioeconomic Status Index (PCA)

The PCA approach is appropriate, but further clarification on the asset selection and weighting would improve reproducibility. Consider adding a brief explanation or table in the supplementary material.

We have added a supplementary file 4 explaining method used to construct the socioeconomic status (line 394).

Discriminant Analysis of Principal Components

Asset that could indicate wealth in the study location was selected based on the previous household survey data and information on locally relevant indicator of wealth (1). We used discriminant analysis of principal components to select assets that best separate SES groups (poorest to richest). The discriminant analysis would maximise the between group variability and minimise within group variability (2). This methods captured heterogeneity in ownership that was not seen using a standard principal component analysis (2).

Result from the Discriminant analysis of Principal Component to construct socio-economic status groupings is shown below (figure is shown in Supplementary File 4).

3. Terminology and Formatting

Please standardize the formatting of parasite names (italicized), units (e.g., g/dL), and terminology (avoid excessive switching between “trichuriasis” and “T. trichiura infection unless contextually justified).

Thank you for pointing this out. We sincerely apologize for missing the suggested formatting in our previous submission and have now corrected accordingly. We have standardized the formatting of parasites names and units throughout the manuscript. We have also carefully reviewed the use of “trichuriasis” and “T. trichiura” infections in the manuscript and have made the relevant changes.

Minor Comments

Line 373: Consider rewording “might indicate gaps in knowledge…” to a more neutral phrase such as “may suggest limited awareness…”.

We have revised the wordings accordingly.

Clarify rationale for the DNA input volume (2 µL) in the qPCR if it differs from standard recommendations.

Although we did not quantify DNA concentrations prior to qPCR, we validated our assay by performing serial dilutions of positive samples. The consistent and proportional Ct shifts confirmed that 2 µL of template did not cause inhibition and was appropriate for reliable amplification. The use of a fixed 2 µL DNA volume is also supported by other qPCR studies targeting different parasites. For example, assays for Strongyloides helminth (3); Plasmodium (4), or Onchocerca spp (5) have all used 2 µL template volumes, often without explicit prior quantification. These examples reflect standard practice in diagnostic qPCR workflows, particularly in field settings.

To improve clarity, we have added the associated references in line 140 in the manuscript draft.

Consider suggesting future work involving multiplex qPCR and infection intensity quantification using more sensitive or quantitative techniques.

We have added a line for future studies suggestion.

Lines 335-357:

“Future studies using multiplex qPCR and more sensitive quantitative techniques could provide important data on the true burden of STH infections in children under five”.

References:

1. Karyana M, Devine A, Kenangalem E, Burdarm L, Poespoprodjo JR, Vemuri R, et al. Treatment-seeking behaviour and associated costs for malaria in Papua, Indonesia. Malar J. 2016;15(1):536.

2. Vyas S, Kumaranayake L. Constructing socio-economic status indices: how to use principal components analysis. Health Policy Plan. 2006;21(6):459-68.

3. Becker SL, Piraisoody N, Kramme S, Marti H, Silue KD, Panning M, et al. Real-time PCR for detection of Strongyloides stercoralis in human stool samples from Cote d'Ivoire: diagnostic accuracy, inter-laboratory comparison and patterns of hookworm co-infection. Acta Trop. 2015;150:210-7.

4. Imwong M, Hanchana S, Malleret B, Renia L, Day NP, Dondorp A, et al. High-throughput ultrasensitive molecular techniques for quantifying low-density malaria parasitemias. J Clin Microbiol. 2014;52(9):3303-9.

5. Mekonnen SA, Beissner M, Saar M, Ali S, Zeynudin A, Tesfaye K, et al. O-5S quantitative real-time PCR: a new diagnostic tool for laboratory confirmation of human onchocerciasis. Parasites & vectors. 2017;10(1):451.

---

## [Decision Letter · Decision Letter 2]

15 Oct 2025

Molecular diagnosis of Trichuris trichiura: Prevalence and associated risk factors in children under five living in a malaria-endemic area in Papua, Indonesia

PONE-D-25-23992R2

Dear Dr. Poespoprodjo,

We’re pleased to inform you that your manuscript has been judged scientifically suitable for publication and will be formally accepted for publication once it meets all outstanding technical requirements.

Kind regards,

Hammed Oladeji Mogaji, Ph.D

Academic Editor

PLOS ONE

Additional Editor Comments (optional):

Reviewers' comments:

Reviewer's Responses to Questions

**Comments to the Author**

Reviewer #1: All comments have been addressed

2. Is the manuscript technically sound, and do the data support the conclusions?

Reviewer #1: Yes

3. Has the statistical analysis been performed appropriately and rigorously?

Reviewer #1: Yes

4. Have the authors made all data underlying the findings in their manuscript fully available?

Reviewer #1: Yes

5. Is the manuscript presented in an intelligible fashion and written in standard English?

Reviewer #1: Yes

Reviewer #1: (No Response)

**Do you want your identity to be public for this peer review?** For information about this choice, including consent withdrawal, please see our Privacy Policy

Reviewer #1: **Yes: ** Uwem Friday Ekpo

---

## [Editor Report · Acceptance letter]

PONE-D-25-23992R2

PLOS ONE

Dear Dr. Poespoprodjo,

I'm pleased to inform you that your manuscript has been deemed suitable for publication in PLOS ONE. Congratulations! Your manuscript is now being handed over to our production team.

Kind regards,

on behalf of

Dr. Hammed Oladeji Mogaji

Academic Editor

PLOS ONE